# Uni-directional ciliary membrane protein trafficking by a cytoplasmic retrograde IFT motor and ciliary ectosome shedding

**Muqing Cao[1†], Jue Ning[1†], Carmen I Hernandez-Lara[1†], Olivier Belzile[1], Qian Wang[1], Susan K Dutcher[2], Yanjie Liu[1], William J Snell[1]***

[1]Department of Cell Biology, University of Texas Southwestern Medical Center, Dallas, United States; [2]Department of Genetics, Washington University, St. Louis, United States

**Abstract** The role of the primary cilium in key signaling pathways depends on dynamic regulation of ciliary membrane protein composition, yet we know little about the motors or membrane events that regulate ciliary membrane protein trafficking in existing organelles. Recently, we showed that cilium-generated signaling in *Chlamydomonas* induced rapid, anterograde IFT-independent, cytoplasmic microtubule-dependent redistribution of the membrane polypeptide, SAG1-C65, from the plasma membrane to the periciliary region and the ciliary membrane. Here, we report that the retrograde IFT motor, cytoplasmic dynein 1b, is required in the cytoplasm for this rapid redistribution. Furthermore, signaling-induced trafficking of SAG1-C65 into cilia is unidirectional and the entire complement of cellular SAG1-C65 is shed during signaling and can be recovered in the form of ciliary ectosomes that retain signal-inducing activity. Thus, during signaling, cells regulate ciliary membrane protein composition through cytoplasmic action of the retrograde IFT motor and shedding of ciliary ectosomes.

***For correspondence:** william. snell@utsouthwestern.edu

[†]These authors contributed equally to this work

**Competing interests:** The authors declare that no competing interests exist.

## Introduction

The primary cilium provides a unique cellular compartment for interaction with the extracellular milieu. Initiated by discoveries in chemosensation in *Caenorhabditis elegans*, of ciliary defects in polycystic kidney disease, and of the central role of cilia in the vertebrate hedgehog (Hh) signaling pathway, we now know that the organelle is a signaling node for multiple pathways in development, disease, and homeostasis (reviewed in *Gerdes et al., 2009*; *Lancaster and Gleeson, 2009*; *Goetz and Anderson, 2010*). Although the signaling functions of the ciliary membrane are crucial for development and homeostasis, we are just beginning to learn the cellular and molecular mechanisms that underlie the dynamic regulation of ciliary membrane protein composition in existing cilia during cilium-generated signaling (*Pazour and Bloodgood, 2008*; *Nachury et al., 2010*; *Hu and Nelson, 2011*; *Malicki and Avidor-Reiss, 2014*).

During ciliogenesis, soluble proteins are delivered into the organelle by intraflagellar transport (IFT) (*Pedersen and Rosenbaum, 2008*). The working model for delivery of ciliary membrane proteins during ciliogenesis is that vesicles derived from the Golgi and containing newly synthesized membrane proteins destined for the cilium traffic via as yet poorly characterized motors to specialized regions of the plasma membrane near the base of the organelle, including the ciliary pocket (*Molla-Herman et al., 2010*; *Rohatgi and Snell, 2010*), and enter the cilium as the organelle elongates. IFT proteins are required for the presence of some membrane proteins in the cilium, but it remains unknown whether IFT is required for their entry (*Mukhopadhyay et al., 2010*; *Wood and Rosenbaum, 2014*). Several proteins and protein

**eLife digest** Nearly every cell in the human body has slender, hair-like structures known as cilia that project outwards from its surface. These structures can sense and respond to light, chemicals and touch, and they are required for normal development. Failure of cilia to form or function in the correct manner can lead to severe diseases—such as kidney disorders, deafness and loss of vision. A major puzzle for researchers who study cilia has been to understand how cells change the composition of these structures as part of their response to a sensory input.

Cilia are ancient structures that were present in early single-celled organisms and researchers interested in cilia have often used a single-celled green alga called *Chlamydomonas reinhardtii* as a model system for their studies. When these algae reproduce sexually, the two types of sex cells sense the presence of each other when their cilia touch and then stick together. This ciliary touching activates signals that are sent into the cells to get them ready to fuse together, much like sperm and egg cells do in animals.

Both ciliary touching and signaling depend on a protein called SAG1, a part of which (known as SAG1-C65) is normally found mostly over the surface membrane of *C. reinhardtii*. Only very small amounts of SAG1-C65 are normally found on cilia; but, when the sex cells' cilia touch, this protein rapidly moves to the end of the cell nearest the cilia via a previously unknown mechanism. SAG1-C65 then becomes much more enriched in the cilia.

Cao, Ning, Hernandez-Lara et al. investigated this process and found that SAG1-C65 movement requires a molecular motor called 'cytoplasmic dynein'. This motor protein typically walks along the inside of cilia to transport other molecules away from the tip and towards the cell membrane. However, Cao, Ning, Hernandez-Lara et al. found that this dynein also carries SAG1-C65 from the membrane of the cells towards the base of the cilia in preparation for it to enter into these structures.

As part of an effort to understand the fate of the protein after it entered cilia, Cao, Ning, Hernandez-Lara et al. discovered that the SAG1-C65 disappeared from the structures without returning to the cell membrane. Instead, SAG1-C65 was packaged within tiny bubble-like structures near the tips of cilia and these packages were then shed from cilia into the external environment. This discovery challenges a widely held view that proteins are only removed from cilia by returning to the cell. Future work will be required to understand more of the molecular details of these processes, which are likely to be present in most cells with cilia.

complexes are involved in trafficking of membrane proteins to the organelle during its formation as well as in establishing the barrier, and their mutational disruption leads to aberrant regulation of constitutively localized ciliary membrane proteins, or to complete absence of cilia and to the ciliopathies (*Garcia-Gonzalo et al., 2011*; *Hildebrandt et al., 2011*; *Sang et al., 2011*; *Williams et al., 2011*; *Chih et al., 2012*).

After the cilium is fully formed, however, the protein composition of the ciliary membrane does not remain fixed, but is dynamically regulated during cilium-generated signaling. In the well-studied cilium-based Hedgehog (Hh) signaling system in vertebrates, the pathway suppressors Ptch and GPR161 are present in the ciliary membrane in the absence of the Hh ligand, and the effector Smoothened (Smo) is mostly excluded from the cilium and present on the cell plasma membrane and in internal vesicles. When Hh binds to it receptor Ptch, Smo becomes enriched in the ciliary membrane where it regulates downstream events in the pathway (*Rohatgi et al., 2007*; *Milenkovic et al., 2009*; *Wang et al., 2009*; *Dorn et al., 2012*; *Mukhopadhyay et al., 2013*). Concomitantly, both Ptch and GPR161 are depleted from the organelle. Although several models have been proposed to account for the fascinating, dynamic, regulated enrichment and loss of Hh effectors in the ciliary membrane, we still know few mechanistic details.

The use of cilia as sensory/signaling organelles is an ancient invention. Interactions between receptors (agglutinin polypeptides) on the cilia of *plus* and *minus* gametes during fertilization in the green alga *Chlamydomonas* trigger an anterograde IFT-dependent signaling pathway within the organelles (*Wang and Snell, 2003*; *Wang et al., 2006*) that activates the gametes for cell–cell fusion (*Snell and Goodenough, 2009*). The *plus* agglutinin polypeptide receptor expressed on *plus* gametes is encoded by the SAG1 gene and the *minus* agglutinin polypeptide receptor on *minus* gametes is

encoded by the SAD1 gene (*Ferris et al., 2005*). In addition to activating the signaling pathway within each type of gamete, interactions between the SAG1 *plus* agglutinin and the SAD1 *minus* agglutinin cause the cilia of the gametes to adhere to each other, thereby bringing the gametes into the close contact required for gamete fusion.

Recently, using *plus* gametes expressing a *SAG1-HA* transgene, we showed that soon after synthesis of the full-length protein encoded by *SAG1*, it is cleaved to yield the N-terminal *plus* agglutinin polypeptide and a C-terminal, integral membrane polypeptide, SAG1-C65 (*Belzile et al., 2013*). We found that although small amounts of SAG1-C65 were on the cilia of resting *plus* gametes, most was excluded from the organelles and present at the plasma membrane. When the cilium-generated signaling pathway was activated, however, the C-terminal SAG1-C65 polypeptide was rapidly recruited to the ciliary membrane through a mechanism that did not require the anterograde IFT motor kinesin 2/FLA10. Moreover, before entering the cilium during signaling, SAG1-C65 became highly polarized, accumulating in the periciliary region as part of a ciliary entry pathway that required cytoplasmic microtubules. Here, we report that during cilium-generated signaling, cells regulate ciliary membrane SAG1-C65 levels by action of the retrograde IFT motor in the cytoplasm and by regulated shedding of SAG1-C65-containing ciliary ectosomes that retain signaling competency and comprise a distinct membrane compartment.

## Results

### The retrograde IFT motor is required for apical polarization and ciliary enrichment of SAG1-C65 during cilium-generated signaling

The presence in *Chlamydomonas* of only a single cytoplasmic dynein, cytoplasmic dynein 1b, and our previous results that cytoplasmic microtubules participated in periciliary accumulation and ciliary entry of SAG1-C65 during signaling raised the possibility that this microtubule minus end-directed IFT motor (*Pazour et al., 1999*) might participate in SAG1-C65 redistribution. The benzoyl dihydroqui-nazolinone, ciliobrevin D, has been shown in metazoans to block cytoplasmic dyneins (*Hyman et al., 2009*; *Firestone et al., 2012*; *Ye et al., 2013*). And, recently *Shih et al. (2013)* showed that ciliobrevin D inhibition of *Chlamydomonas* cytoplasmic dynein 1b (DHC1b) strongly reduced retrograde IFT. We tested for a role of the retrograde IFT motor in SAG1-C65 redistribution during signaling using ciliobrevin D. Early during ciliary adhesion and cilium-generated signaling, activation of a ciliary adenylyl cyclase leads to an ~15-fold increase in cellular cAMP that activates gametes to prepare for fusion. Thus, it is possible to study cellular events activated by the signaling pathway, such as redistribution of the agglutinin polypeptide, release of cell walls, and upregulation of transcripts for gamete-specific proteins, in gametes of a single mating type by incubating them in the cell-permeable analogue, db-cAMP (*Pijst et al., 1984b*; *Pasquale and Goodenough, 1987*; *Goodenough, 1989*; *Hunnicutt et al., 1990*; *Belzile et al., 2013*; *Ning et al., 2013*). We incubated *SAG1-HA/sag1-5* gametes (which express a tagged SAG1-C65 polypeptide, SAG1-C65-HA) (*Belzile et al., 2013*) with and without ciliobrevin D for 20 min, activated them by addition of db-cAMP for 5 min in the continued presence of the inhibitor, and then assessed SAG1-C65-HA localization.

As shown previously (*Belzile et al., 2013*), whereas SAG1-C65-HA showed apical localization in only a small portion of resting gametes, the protein became apically localized after gametes were activated by incubation in db-cAMP for 5 min (*Figure 1A,B*). Incubation of resting cells in ciliobrevin D reduced the already low percentage of cells with apically localized SAG1-C65-HA (*Figure 1B*), and inhibited SAG1-C65-HA redistribution to the apical ends in cells incubated for 5 min in db-cAMP (*Figure 1A,B*). Consistent with the results of *Shih et al. (2013)* that ciliobrevin did not completely inhibit the motor activity of cytoplasmic dynein-1b, after longer incubation in db-cAMP, SAG1-C65-HA redistribution returned (not shown). These results indicated that the retrograde IFT motor was essential for the rapid redistribution of SAG1-C65-HA to the peri-ciliary region at the cell apex.

Ciliobrevin D inhibition of signaling-induced SAG1-C65-HA apical localization also resulted in a concomitant inhibition of accumulation of the protein in the cilia. Cilia of resting gametes contained little SAG1-C65-HA as assessed by both IF and immunoblotting of isolated cilia, but upon 5-min incubation in db-cAMP, SAG1-C65-HA in the organelles increased in control, but not in the ciliobrevin samples (*Figure 1A,C*, upper panels). The effects of ciliobrevin D were transient, and cells washed out of the inhibitor rapidly regained the ability to respond to db-cAMP by accumulating SAG1-C65-HA in their cilia (*Figure 1C*). IF analysis showed that the array of cytoplasmic microtubules was unaffected by

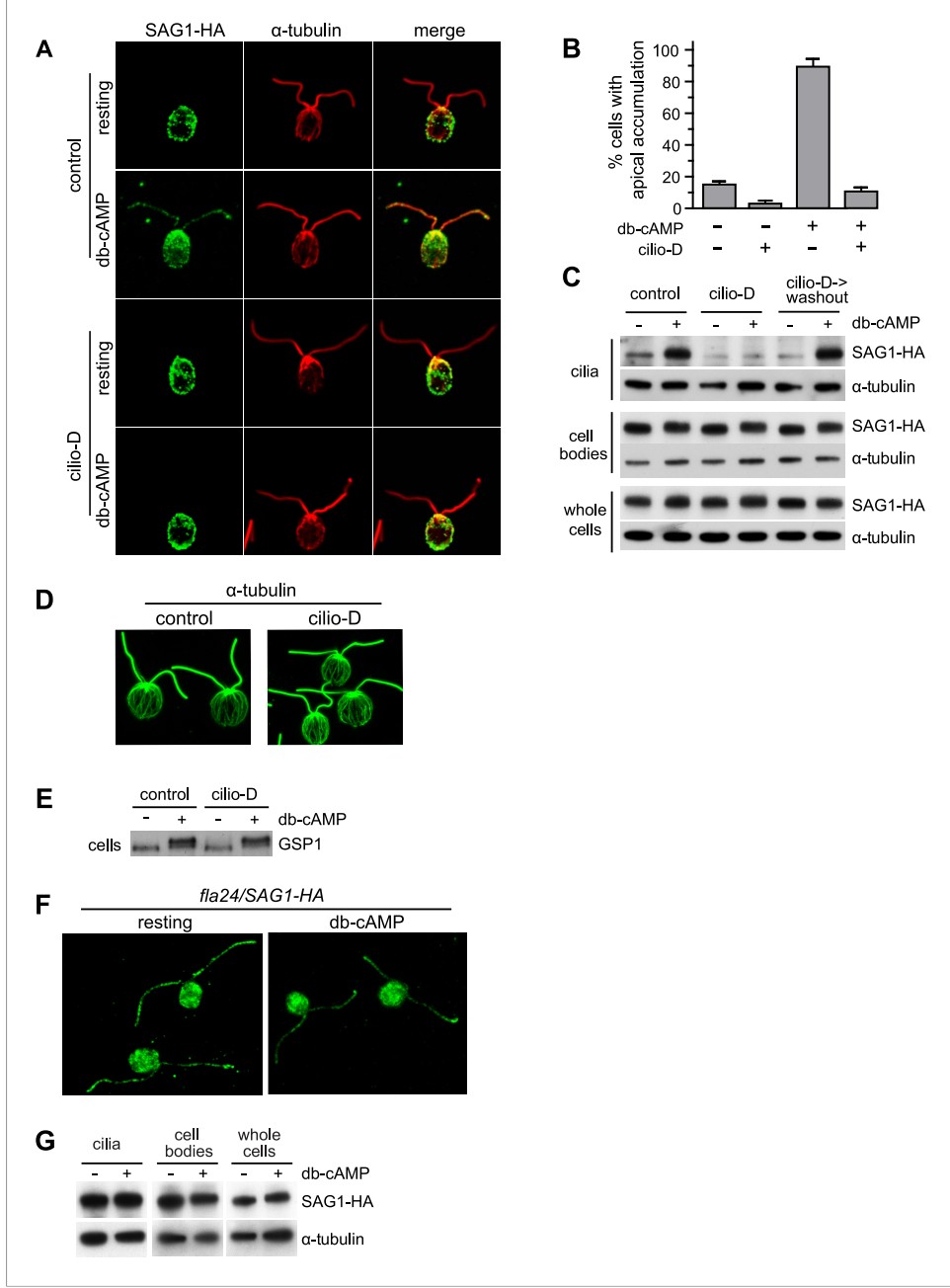

**Figure 1**. Cytoplasmic dynein 1b is required for the rapid, signaling-induced apical localization and ciliary enrichment of SAG1-C65-HA. (**A**) Confocal images of resting gametes in the presence and absence of Cilio-D for 20 min and gametes activated for 5 min in the presence and absence of Cilio-D. (**B**) Percent of cells showing apical localization of SAG1-HA (see 'Materials and methods'). Data are from three experiments. 100 cells were scored for each condition. Error bars indicate ± SD. (**C**) Immunoblots of cilia, cell bodies and whole cells (8 µg protein/lane) of SAG1-HA cells treated as indicated and activated (or not) with db-cAMP for 5 min. (**D**) Confocal images showing cytoplasmic microtubules in control and Cilio-D treated plus gametes. (**E**) Immunoblot for GSP1 of *plus* cells incubated with or without Cilio-D and with or without db-cAMP. (**F**) Confocal images of resting and db-cAMP activated gametes showing high levels of SAG1-C65-HA in cilia of resting *fla24* gametes and lack of redistribution upon activation. (**G**) Immunoblots of resting and activated whole cells, cell bodies, and cilia (2.5 µg protein/lane) of *fla24* gametes.

ciliobrevin D (*Figure 1D*). And, the inhibitor had no effect on total cellular SAG1-C65-HA (*Figure 1C*, lower panels). Interestingly, and consistent with the IF results that ciliobrevin D treatment of resting cells reduced the percent with apical localization of SAG1-C65-HA, the basal level of SAG1-C65-HA in cilia isolated from ciliobrevin D-treated resting gametes was lower than in the control, resting gametes (*Figure 1C*). We also examined whether the inhibitor blocked signaling per se, by testing its affects on signaling-induced phosphorylation of the homeodomain protein, GSP1 (*Wilson et al., 1999*). As shown in *Figure 1E*, gametes in ciliobrevin D remained capable of responding to db-cAMP as assessed by the phosphorylation-related change in migration of GSP1 in immunoblots (*Wilson et al., 1999*). Thus, apical localization and ciliary enrichment of SAG1-C65 during signaling were both inhibited by the cytoplasmic dynein inhibitor, ciliobrevin D, indicating that the retrograde IFT motor acts in the cytoplasm to enrich SAG1-C65 in the peri-ciliary region concomitant with entry of the protein into the ciliary membrane.

As a further test for a role of the retrograde IFT motor in SAG1-C65 trafficking, we introduced the SAG1-HA transgene into the cytoplasmic dynein mutant strain *fla24*, which has an L3242P mutation in dynein heavy chain 1b (DHC1b) (*Lin et al., 2013*). *fla24* exhibits a temperature-sensitive ciliary phenotype, being ciliated at 21°C, and without cilia after 4–6 hr at 32°C. Even at 21°C, however, DHC1b and its associated light chain protein D1bLIC are reduced dramatically and the level of D1bLIC in *fla24* cells is less than 7% of wild type at 21°C (*Lin et al., 2013*). Thus, we tested the effects of reduced retrograde IFT motor function on SAG1-C65-HA apical localization and ciliary entry by using *SAG1-HA/fla24* gametes at 21°C. Consistent with the results with ciliobrevin D, at 21°C, SAG1-C65-HA failed to undergo rapid apical redistribution in the DHC1b-depleted *fla24* gametes incubated in db-cAMP (*Figure 1F*) and the amount of SAG1-C65-HA in cilia did not increase (*Figure 1G*). Interestingly, the depleted levels of the retrograde IFT motor in the *fla24* cells were also associated with aberrantly large amounts of ciliary SAG1-C65-HA in resting gametes (*Figure 1F,G*). Thus, the retrograde IFT motor is required for trafficking of SAG1-C65-HA during ciliogenesis of gametes, and it is required for redistribution of SAG1-C65-HA to the peri-ciliary region and into the cilium during cilium-generated signaling.

## SAG1-C65-HA is depleted from cells during sustained ciliary adhesion and signaling

We also followed the fate of SAG1-C65-HA when SAG1-C65-HA *plus* gametes were mixed with fusion-defective *hap2 minus* gametes and allowed to undergo sustained ciliary receptor-induced signaling (*Liu et al., 2008*). As expected, cilia isolated after 30 min of signaling possessed substantial amounts of SAG1-C65-HA (*Belzile et al., 2013*), and both the ciliary and cell body SAG1-C65-HA levels remained constant for 3 hr (*Figure 2A*, control). Previously (*Ning et al., 2013*), we showed that gametes upregulate SAG1 transcripts fourfold to fivefold during gamete activation (*Figure 3B*), and thus we were surprised that the protein levels failed to increase during the 3-hr experiment. Although protein levels do not necessarily reflect transcript levels, both the transcript (*Ning et al., 2013*) and protein amounts of the *minus* gamete-specific membrane protein HAP2 increased during gamete activation (*Figure 2B,C*). Therefore, we tested whether SAG1-C65-HA was being synthesized and then turned over by the gametes undergoing sustained, agglutinin receptor-induced ciliary signaling by examining protein amounts in samples in which protein synthesis was blocked. At 30 min after mixing the gametes together in the protein synthesis inhibitor, cycloheximide (CH), the amount of SAG1-C65-HA in the cilia was similar to that of control, signaling gametes as shown previously (*Belzile et al., 2013*). At 1.5 hr, however, SAG1-C65-HA in the cilia of the CH-treated samples had decreased, and at 3 hr the amount of SAG1-C65-HA in the cilia had fallen dramatically (*Figure 2A*, CH).

Notably, SAG1-C65-HA was also depleted from the cell body during agglutinin receptor-induced signaling in CH (*Figure 2A*, CH). This wholescale depletion of cellular SAG1-C65-HA in CH was not simply a reflection of normal turnover, because the amount of SAG1-C65-HA in resting gametes was unchanged after 3 hr in CH (*Figure 2D*). Moreover, SAG1-C65-HA also remained constant in gametes activated with db-cAMP in the presence of CH (*Figure 2D*), and no SAG1-C65-HA could be detected in the medium (not shown). And, gametes undergoing sustained agglutinin-induced signaling in CH in the presence of the protein kinase inhibitor staurosporine, which blocks recruitment of SAG1-C65-HA to cilia (*Belzile et al., 2013*), also did not lose SAG1-C65, indicating that interactions of the agglutinin receptors on *plus* and *minus* gametes and signaling were required for loss (*Figure 2D*).

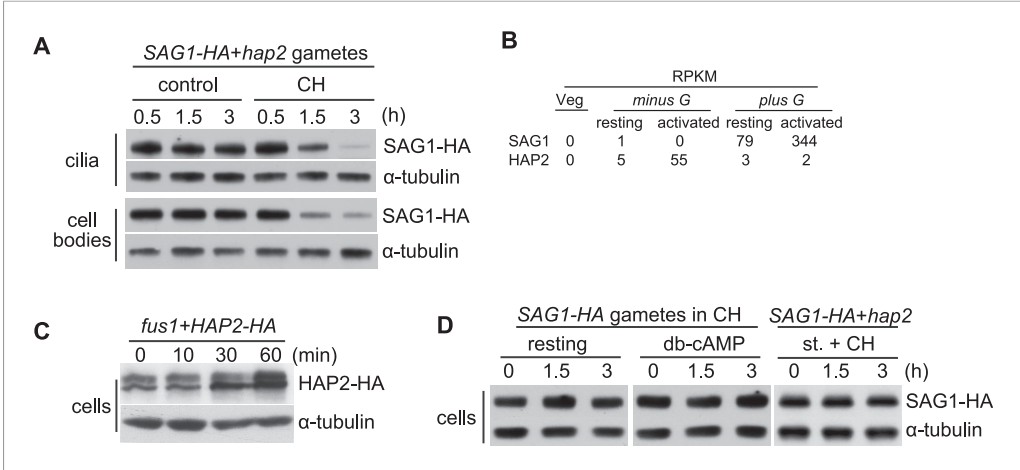

**Figure 2**. SAG1-C65-HA is lost from cells during adhesion and signaling. (**A**) Immunoblot analysis of SAG1-C65-HA in isolated cilia and cell bodies (8 µg protein/lane) from cells activated by mixing with *hap2 minus* gametes. Cells were pretreated (or not) for 15 min with 10 mg/ml cycloheximide (CH) before mixing. (**B**) RNA-Seq data from **Ning et al. (2013)** showing transcript abundance as median reads per kilobase per million mapped reads (RPKM) values of SAG1 and HAP2 transcripts from vegetative cells, resting gametes, and activated gametes of both mating types. (**C**) Immunoblots using HA and tubulin antibodies of HAP2-HA *minus* gametes ($1.5 \times 10^7$ cells/ml) mixed with an equal number of *fus1 plus* gametes for the indicated times. (**D**) Immunoblot analysis of SAG1-C65-HA in isolated SAG1-HA resting cells, SAG1-HA cells activated with db-cAMP, and SAG1-HA cells activated by undergoing ciliary adhesion with *hap2 minus* gametes in the presence of the protein kinase inhibitor staurosporine (1 µM; st.).

## SAG1-C65-HA is released into the medium during adhesion and signaling

To determine whether the lost SAG1-C65-HA had been shed into the medium, we activated sustained agglutinin receptor interactions by mixing SAG1-C65-HA *plus* gametes and fusion-defective *minus* gametes together for 3 hr, harvested the cells, and then determined SAG1-C65-HA amounts in equal portions of the cells and the medium. As shown in *Figure 3A* (upper panels), cells indeed released SAG1-C65-HA into the medium during receptor-activated signaling. Moreover, the amount recovered in the medium after 3 hr was three times as much as present in the starting cells (*Figure 3A,B*), indicating that the relatively constant levels of cell body and ciliary SAG1-C65-HA during sustained agglutinin receptor interactions and signaling reflected a dynamic balance between synthesis, ciliary delivery, and release from the cilia. Moreover, when the cells were mixed together in the presence of the protein synthesis inhibitor cycloheximide, nearly 75% of the SAG1-HA in the cells at T = 0 was recovered in the medium after 3 hr of receptor-activated signaling, demonstrating that release was the major fate of the protein (*Figure 3A*, lower panels and *Figure 3B*). Consistent with the results above, SAG1-C65-HA was absent from the medium of adhering gametes in which signaling and ciliary recruitment of SAG1-C65-HA were blocked by the addition of staurosporine (*Figure 3C*). On the other hand, compared to control *SAG1-HA* gametes undergoing receptor-activated signaling induced by mixing them with *hap2 minus* gametes, *SAG1-HA* gametes that had been activated by pre-incubating them with db-cAMP and then mixed with *hap2* gametes, released increased amounts of SAG1-C65-HA into the medium, even when staurosporine was also added to the medium (*Figure 3C*). Thus, once gametes had been activated with db-cAMP, staurosporine no longer blocked release of SAG1-C65-HA during agglutinin receptor interactions.

Differential centrifugation showed that the SAG1-C65-HA in the medium was particulate. It was in the supernatant after sedimenting the cells at low speed, remained soluble after centrifugation at intermediate speed (20,000×*g*), and was sedimented upon centrifugation at 200,000×*g* (*Figure 3D*). Thus, nearly all of the SAG1-C65-HA that was lost during ciliary adhesion was recovered in the medium in a particulate form. Biochemical fractionation indicated that SAG1-C65-HA harvested from the medium by high-speed centrifugation indeed was an integral membrane protein (*Belzile et al., 2013*), A high salt wash failed to release the protein, but release required detergent (*Figure 3E*).

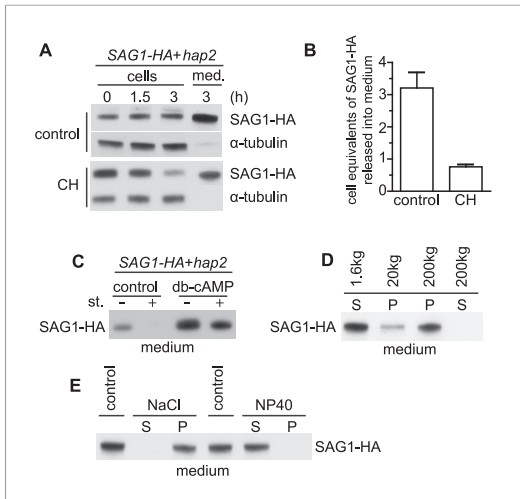

**Figure 3**. Adhering gametes release SAG1-C65-HA in a membrane-associated form. (**A**) Immunoblot analysis of SAG1-C65-HA in cells and medium harvested after undergoing adhesion and signaling with fusion-defective *hap2 minus* gametes for the indicated times. 1 × 10⁶ cell equivalents were loaded in each lane. (**B**) Number of cell equivalents of SAG1-HA released into the medium 3 hr after mixing SAG1-C65-HA gametes with *hap2* gametes in the presence and absence of CH as assessed by quantitative immunoblotting. Results are averages from the experiment shown in (**A**) and at least one other independent experiment. Error bars show S.D. (**C**) Immunoblot analysis of SAG1-C65-HA in the medium collected from SAG1-HA gametes undergoing adhesion with *hap2 minus* gametes. Before mixing, cells were activated (or not) with db-cAMP and then mixed together in the presence or absence of 1 μM staurosporine (st.) for 3 hr. (**D**) Immunoblot analysis of SAG1-HA distribution after differential centrifugation of medium from SAG1-C65-HA and *hap2* gametes that had been adhering for 3 hr. S, supernatant; P, pellet. (**E**) Distribution of SAG1-C65-HA after 200,000×*g* centrifugation of ectosome samples that had been incubated on ice for 20 min in 0.5 M NaCl or 2% NP-40 in HMDEK buffer. S, supernatant; P, pellet.

Taken together, these results indicated that ciliary receptor interactions and signaling triggered release of SAG1-C65-HA from the cilia into the medium in a particulate, membrane-associated form.

## Ciliary receptor interactions and signaling induce shedding of SAG1-C65-containing vesicles from cilia

Several groups have reported that vegetative cells and gametes constitutively release membrane vesicles (*Wiese, 1965*; *Bergman et al., 1975*; *Snell, 1976*; *Dentler, 2013*), and very recently *Wood et al. (2013)* demonstrated release of biologically active ciliary ectosomes from vegetative cells during cell division. Moreover, Goodenough and Heuser reported several years ago that cilia on adhering gametes possessed associated membrane vesicles that were absent from non-adhering samples (*Goodenough and Heuser, 1999*). Those studies did not determine whether the vesicles seen in the samples of the mixed gametes represented vesicles that were in the medium of each gamete type before the cells were mixed, or if their formation was induced by adhesion.

We washed gametes into fresh medium to remove any vesicles that might have been present in the medium, and used negative staining and TEM to examine the cilia of non-adhering and adhering gametes. As expected, the cilia of the non-adhering gametes exhibited a smooth membrane with no evidence of particulate material (*Figure 4A*). The surfaces of cilia of *plus* and *minus* gametes that had been washed into fresh medium and mixed together for 10–15 min, however, were strikingly different (*Figure 4B*). Adhesion had triggered appearance of vesicles on the ciliary surfaces, some along the shafts of the organelles, and many near the ciliary tips. Some of the vesicles were in clusters, but many were single vesicles in intimate association with the ciliary membrane. Formation of vesicles during ciliary adhesion was unique to cilia, and we did not observe vesicles associated with the cell body surface of the gametes. Importantly, similar to the release of SAG1-C65-HA from cilia detected by biochemical methods, formation of the vesicles required ciliary receptor-induced signaling and vesicles were not detected on cilia of *plus* gametes alone activated by db-cAMP (*Figure 4D*). Taken together, the simplest explanation for these results is that ciliary adhesion and signaling triggered shedding of ciliary membranes in the form of ciliary ectosomes.

## Ciliary ectosomes represent a unique ciliary membrane compartment and possess ciliary signaling activity

Negative staining TEM confirmed that the particulate material isolated from the medium of adhering gametes indeed was composed of vesicles similar to those being released from the surface of the adhering cilia (*Figure 4E*). To our surprise, the protein composition of these ciliary ectosomes was

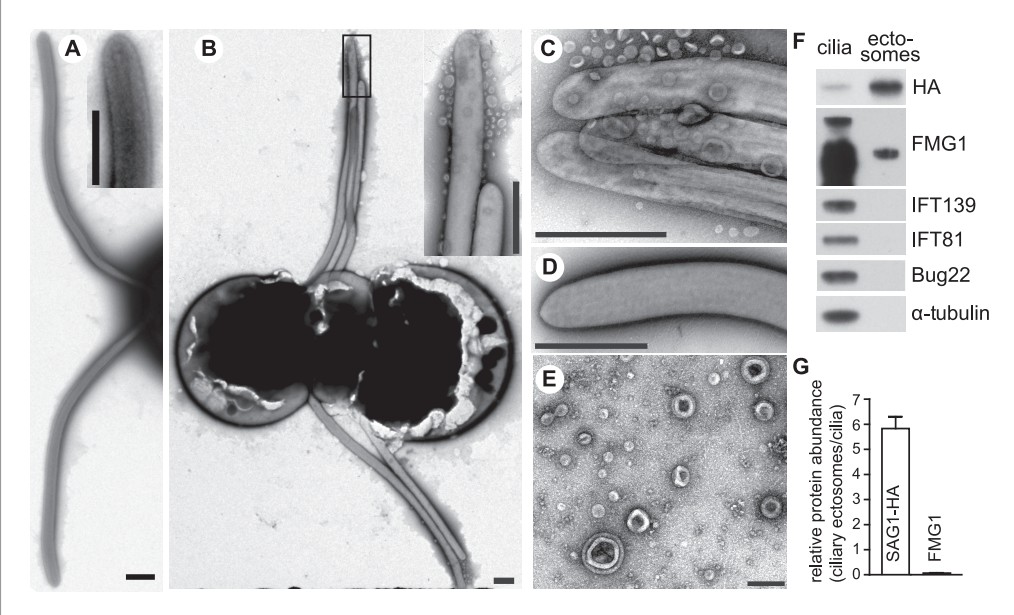

**Figure 4**. Adhering gametes release SAG1-C65-HA as ciliary ectosomes. (**A**) Negative stain transmission electron micrographs of resting wild type *plus* gametes showing the smooth membranes of the cilia. The inset shows a higher magnification view of the tip of a cilium. (The scale bars shown in this and subsequent TEM images are 1000 nm, with the exception that the bar in *Figure 4E* is 200 nm.) (**B**) Wild type *plus* and *hap2 minus* gametes were washed into fresh N-free, mixed together for 10–15 min, and prepared for TEM. The adhering cilia contained larger numbers of associated vesicles. The inset shows a higher magnification view of the vesicles at the ciliary tips. (**C**) High magnification view of adhering cilia of wild type *plus* and *hap2 minus* gametes showing vesicles. (**D**) Vesicles were absent from the cilia of db-cAMP-activated wild type *plus* gametes. (**E**) TEM of ciliary ectosomes harvested from the medium of adhering wild type *plus* and *hap2 minus* gametes. (**F**) Immunoblots with the indicated antibodies of equal protein amounts (8 µg/lane) of ciliary ectosomes (right lane) and cilia isolated from a mixture of adhering SAG1-C65-HA *plus* gametes and *hap2 minus* gametes (left lane). (**G**) Relative abundance of SAG1-HA and FMG1 in equal amounts of protein of ciliary ectosomes compared to cilia in a representative experiment. Error bars show S.D. from at least three quantifications of immunoblots.

The following figure supplement is available for figure 4:

**Figure supplement 1**. Model illustrating unidirectional trafficking of SAG1 during ciliary adhesion and signaling.

much different than that of isolated cilia. Immunoblotting (*Figure 4F*) showed that isolated cilia contained typical ciliary proteins including IFT81, IFT139, alpha tubulin, Bug22 (an axonemal protein; *Meng et al., 2014*), SAG1-C65, and the major ciliary membrane protein, FMG1. On the other hand, immunoblot analysis of an equal amount of ciliary ectosome protein showed that SAG1-C65-HA was enriched nearly sixfold in ciliary ectosomes compared to cilia, but most other ciliary proteins were de-enriched (*Figure 4F,G*). FMG1 was detectable, but strikingly de-enriched (relative abundance <0.1 in ciiary ectosomes compared to cilia; *Figure 4F,G*), and IFT81, IFT139, Bug22, and alpha tubulin typically were not detectable. Thus, SAG1-C65-containing ciliary ectosomes represented a unique ciliary compartment released from cilia during receptor-activated signaling.

We examined whether the isolated ectosomes contained biological activity by mixing them with *minus* gametes and testing for agglutinin receptor interactions and gamete activation. As shown by TEM (*Figure 5A*, right panel), the vesicles adhered along the lengths of the cilia of the *minus* gametes, demonstrating that the ectosomes possessed the *plus* agglutinin polypeptide that bound to the SAD1 agglutinin polypeptide on the cilia of the *minus* gametes. Vesicles did not adhere to flagella of *minus* vegetative cells, which do not express the *minus* agglutinin, demonstrating that binding was specific (*Figure 5A*, left panel). A direct binding assay confirmed the TEM results. Isolated ectosomes were incubated with *minus* vegetative cells or *minus* gametes for 15 min, followed by harvesting of the cells

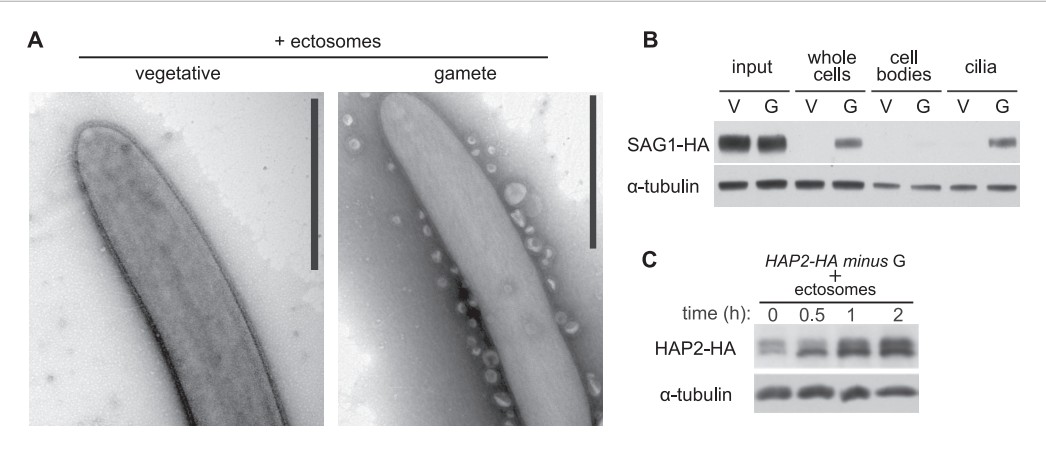

**Figure 5**. Ciliary ectosomes possess agglutinin receptor binding and signaling activity. (**A**) TEM of cilia on *minus* vegetative cells (left panel) and gametes (right panel) that had been mixed with ciliary ectosomes. Only the cilia on the gametes bound ciliary ectosomes. (**B**) Immunoblot analysis of binding of SAG1-C65-HA-containing ectosomes to whole cells, cell bodies, and cilia of *minus* gametes. The initial sample of cells + ectosomes is also shown (input). The lanes contain equal cell equivalents. (**C**) Cilium-generated signaling was activated when ciliary ectosomes were mixed with *HAP2-HA minus* gametes, as assessed by activation-induced upregulation of expression of the gamete fusion protein, HAP2.

by centrifugation. Analysis of the harvested whole cells by immunoblotting showed that only *minus* gametes bound the SAG1-C65-HA vesicles. Furthermore, cell fractionation showed that the vesicles bound only to the cilia, and little if any binding was detected in the cell body fraction (*Figure 5B*).

Finally, we tested whether the isolated ectosomes possessed the ability to activate signaling in *minus* gametes. When *minus* gametes are activated by mixing with *plus* gametes, they upregulate synthesis of several genes specifically expressed in *minus* gametes, including the *minus* gamete fusion protein HAP2 (*Liu et al., 2010*; *Ning et al., 2013*). We tested for signaling activity of the ectosomes by mixing them with *minus* gametes expressing a tagged form of the gamete fusion protein HAP2 (HAP2-HA) and (as a control) with *minus* vegetative cells, and assessed HAP2-HA protein levels. As shown in *Figure 5C*, addition of the ectosomes indeed led to substantial increase in HAP2-HA. Thus, the isolated ciliary ectosomes were capable of activating cilium-generated signaling.

## Discussion

We report here that membrane protein trafficking through the cilium during cilium-generated signaling in *Chlamydomonas* is uni-directional and depends on the concerted action of two cellular processes, protein delivery to the ciliary base by the retrograde IFT motor, cytoplasmic dynein 1b, and shedding of a unique ciliary membrane compartment in the form of ciliary ectosomes. Use of both a retrograde IFT motor mutant, *fla24* and the cytoplasmic dynein 1b inhibitor, ciliobrevin D, showed that the signaling-triggered rapid polarization of SAG1-C65-HA and its entry into the ciliary membrane were inhibited when motor function was impaired (*Figure 1*). Thus, the retrograde IFT motor transports SAG1-C65-HA from distal cellular membrane sites to the periciliary region, either within the plasma membrane or as vesicles within the cytoplasm. The increased amounts of SAG1-C65-HA in the cilia of resting *fla24* gametes (*Figure 1F,G*) also indicated that the retrograde motor is important for membrane protein trafficking during the ciliogenesis that accompanies gamete formation. In their studies of a conditional DHC1b mutant, *dhc1b-3*, *Engel et al. (2012)* found several differences in ciliary protein composition compared to wild-type. Whether the increased amount of SAG1-C65-HA in the *fla24* cilia reflects a direct or indirect role for the retrograde motor in regulating ciliary membrane protein composition is unclear (*Kim et al., 2009*; *Ocbina et al., 2011*). Although several membrane-associated proteins and protein complexes are strongly implicated in targeting ciliary membrane proteins to the periciliary region in many cell types (*Nachury et al., 2010*; *Wood and Rosenbaum, 2014*), the motors that carry ciliary-destined membrane proteins to the base of the organelle

have been largely unidentified. One notable exception is photoreceptor cells in which cytoplasmic dynein 1 (not the retrograde IFT motor) is involved in trafficking of rod outer segment membrane proteins from the golgi to the base of the connecting cilium (*Tai et al., 1999*; *Kong et al., 2013*).

*Kim et al. (2009)* reported that treatment of mammalian cells with vinblastine, an agent that disrupts cytoplasmic microtubules, had no effect on Smoothened accumulation in the primary cilium many hours after addition of the Hh ligand. On the other hand, the parent compound of ciliobrevin D, HPI–4, inhibited Hh-induced ciliary accumulation of Smoothened by 60–70% 12 hr after addition of the Hh ligand, raising the possibility that a cytoplasmic dynein is involved in trafficking of Smoothened to the cilium in metazoans (*Hyman et al., 2009*). And *Ye et al. (2013)* found that ciliobrevin unexpectedly inhibited anterograde IFT after a 30 min incubation, leading them to suggest that a cytoplasmic dynein might be involved in the delivery of IFT complexes in the cytoplasm to the base of the cilium.

In addition to this signaling-regulated delivery of pre-existing membrane proteins to an existing organelle, cells also deliver pre-existing and newly synthesized membrane proteins to the cilium during ciliogenesis (*Rosenbaum et al., 1969*; *Wood and Rosenbaum, 2014*). And, vectorially labeled proteins on the membrane of de-ciliated cells become incorporated into cilia as they regrow (*Hunnicutt et al., 1990*; *Dentler, 2013*). It will be interesting to determine whether ciliary growth in *Chlamydomonas* depends on cargo transport along cytoplasmic microtubules by cytoplasmic dynein 1b. Given that growth of new cilia is dependent on anterograde IFT, whereas signaling-induced movement of pre-existing SAG1-C65-HA into existing cilia is independent of anterograde IFT, it may be that cells use one membrane protein trafficking modality for ciliary assembly and a different one for regulated ciliary enrichment of pre-existing membrane proteins.

Studies many years ago showed that cilia of *Chlamydomonas* vegetative cells and gametes undergo constitutive membrane shedding (*Wiese, 1965*; *Bergman et al., 1975*; *Snell, 1976*) of vesicles similar in protein composition to the ciliary membrane. The recent work of *Dentler (2013)* showed that vegetative cells shed over 15% of their ciliary membrane per hr, and that the protein composition of the constitutively shed membrane vesicles was similar to that of the ciliary membrane per se. In our current studies, however, resting gametes did not shed SAG1-C65-containing ciliary ectosomes, but shedding of such ectosomes required receptor interactions and signaling. It is likely that the vesicles *Goodenough and Heuser (1999)* observed 25 years ago on adhering cilia corresponded to those we isolated from the medium. Notably, the shed vesicles we isolated were enriched in SAG1-C65-HA and de-enriched in the 350 kDa major membrane protein compared to cilia. And, nearly the entire cellular complement of pre-existing SAG1-C65-HA was shed in ~3 hr. Thus, receptor signaling-induced shedding of SAG1-C65-containing ciliary ectosomes was much more rapid and selective than the previously reported, constitutive shedding of ciliary membrane vesicles, and likely depends on a mechanism distinct from that used for constitutive shedding.

Our discovery that the isolated ciliary ectosomes possessed the ability to bind to the cilia of *minus* gametes and activate cilium-generated signaling (*Figure 5*) indicated that the ectosomes also contained the *plus* receptor agglutinin polypeptide, which likely is an N-terminal fragment of the 340 kDa full length protein encoded by the *SAG1* gene (*Belzile et al., 2013*). Several workers have reported that the agglutinin activity is lost during adhesion (*Weise and Wiese, 1978*; *Snell and Moore, 1980*; *Adair et al., 1983*; *Pijst et al., 1984a*; *Hunnicutt et al., 1990*; *Hunnicutt and Snell, 1991*), and, using a semi-quantitiative bioassay, we reported that over 60% of it could be recovered in an uncharacterized form in the medium (*Hunnicutt et al., 1990*). In future studies, it will be interesting to determine if the integral membrane polypeptide SAG1-C65 is the membrane anchor for the agglutinin polypeptide, which lacks predicted transmembrane domains. These considerations lead to the model that release of ciliary ectosomes during ciliary adhesion and signaling underlies the dynamic nature of ciliary adhesion evidenced by constant cilia adhesion and de-adhesion within mixtures of *plus* and *minus* gametes. From another perspective, because the released ectosomes retain biological activity, their formation might help to maximize the probability of zygote formation.

As indicated above, recent studies showed that regulated formation of ciliary ectosomes is not unique to the cilia of gametes or possibly not even to *Chlamydomonas*. *Wood et al. (2013)* reported that *Chlamydomonas* vegetative cells release ciliary ectosomes that contain an enzyme required for release of the extracellular matrix during cell division (*Kubo et al., 2009*). And, several groups have identified biologically active microvesicles from metazoans that contain ciliary proteins and are capable of binding to cilia (*Hogan et al., 2009*; *Bakeberg et al., 2011*; *Chacon-Heszele et al., 2014*). Regulating the membrane protein composition of the cilium is a central feature of cilium-dependent

signaling. Current models posit that regulation of ciliary membrane protein composition is controlled by bidirectional flow of membrane proteins from the cell body to the cilium and back. Direct evidence has been lacking, however, that ciliary membrane proteins lost from the cilium during signaling actually return to the cell, because of the difficulty in determining the fate of ciliary proteins in most systems. Our results demonstrate that cells can maintain a steady-state level of a ciliary membrane protein by unidirectional flow (*Figure 4—figure supplement 1*) in which proteins move from the cell body to the cilium and then into the extracellular milieu.

## Materials and methods

### Strains and special chemicals

*Chlamydomonas reinhardtii* strains *21gr* (mating type *plus*; CC-1690), *6145C* (mating type *minus*; CC-1691), and the fusion-defective *minus* strain *hap2* are available from the *Chlamydomonas* Genetics Center, University of Minnesota, St. Paul. A *SAG1-HA* transgene was introduced into the ciliary adhesion mutant *mt+/sag1-5* as previously described (*Belzile et al., 2013*). Staurosporine, cycloheximide, dibutyryl-cAMP, and poly-l-lysine were from Sigma–Aldrich (St. Louis, MO) and ciliobrevin D was from Calbiochem (Darmstadt, Germany). Dhc1b mutant *fla24 plus* cells expressing *SAG1-HA* were obtained from a cross between *fla24 minus* gametes and SAG1-HA/*sag1-5 plus* gametes.

### Cell culture, cell fractionation, and harvesting of ciliary ectosomes

Cell growth, induction of gametogenesis by transfer of vegetatively growing cells into N-free medium, fractionation of cells into cell bodies and cilia, and storage of samples were as previously described (*Belzile et al., 2013*). Ciliary ectosomes were isolated from the medium as follows. The indicated *plus* and *minus* gametes were pre-treated with lysin for 30 min to remove cell walls (*Hunnicutt et al., 1990*). After washing with fresh N-free medium, the *plus* and *minus* gametes were mixed with aeration for the times indicated in the figure legends. The samples were centrifuged at 1600×$g$ for 10 min to sediment the cells. The supernatant medium was centrifuged at 20,000×$g$ for 30 min and the resulting supernatant was centrifuged at 200,000×$g$ for 60 min in a TLA 100.3 rotor (Beckman) to sediment membrane vesicles, which were resuspended in N-free medium or HMDEK (20 mM pH = 7.2 HEPES, 5 mM $MgCl_2$, 1 mM dithiothreitol, 1 mM EDTA, 25 mM KCl) buffer.

### Ciliary ectosome binding assay

Ciliary ectosomes harvested from $1.2 \times 10^8$ adhering gametes were mixed with freshly washed *minus* gametes ($4 \times 10^7$ cells) in 0.4 ml N-free medium with gentle agitation. After 15 min, after removing a portion to be used as input, a portion was centrifuged at 20,000×$g$ for 2 min to sediment whole cells and any bound vesicles. The remaining portion was de-ciliated, and centrifuged at 600×$g$ for 2 min to sediment the cell bodies. The supernatant was centrifuged at 20,000×$g$ for 2 min to sediment the cilia. All the fractions were boiled in 1 × SDS sample buffer for 5 min, and then subjected to SDS/PAGE analysis on 4–20% gradient gels (GenScript, USA).

### SDS-PAGE and immunoblotting

SDS-PAGE and immunoblotting were essentially as described previously (*Cao et al., 2013*) with slight modifications (*Belzile et al., 2013*). Cells, cilia, and ciliary ectosome samples were boiled in 1 × SDS sample buffer for 5 min, and then subjected to SDS-PAGE analysis on 4–20% gradient gels (GenScript, USA). The antibodies used for immunoblotting were anti-HA (1:1000; Roche), anti-α-tubulin (1:3000 or 1:200,000; Sigma), anti-GSP1 (1:20,000; *Wilson et al., 1999*), anti-FMG1 (1:100,00), anti-IFT139 (1:50,000), anti-IFT81 (1:1000), and anti-BUG22 (1:500,000). Mouse monoclonal antibody against FMG1 was generously provided by Robert Bloodgood (University of Virginia). Monoclonal antibodies against IFT81 and IFT139 were generously provided by Dennis Deiner and Joel Rosenbaum (Yale University). Rabbit polyclonal antibody against BUG22 was generously provided by Dan Meng and Junmin Pan (Tsinghua University). The amount of SAG1-HA released into the medium during adhesion was determined by use of ImageJ analysis of immunoblots of equal proportions of cells and medium. Determinations were made under conditions in which the amount of sample loaded was linear with the signal obtained. Similar methods were used to determine the relative abundance of FMG1 and SAG1-HA in equal protein amounts of ciliary ectosomes compared to cilia.

## Immunofluorescence microscopy

Immunofluorescence was carried out essentially as described previously (Snell, 1976; Belzile et al., 2013). In some cases, the protocol was modified as follows (Cao et al., 2013): Cells in N-free medium were collected by centrifugation, resuspended in N-free medium, and fixed for 2 min at room temperature in 4% paraformaldehyde in N-free medium. After removing the fixation buffer, the cells were resuspended in PBS and allowed to adhere to 0.1% poly-l-lysine-coated microscope slides for 10 min at room temperature. The slides were immersed in ice-cold 100% methanol for 10 min at –20°C, removed, and allowed to air dry. The primary antibodies were anti–α-tubulin (1:400; Sigma) and rat anti-HA (1:100; Roche). The secondary antibodies were Alexa Fluor 488 goat anti-rat/mouse IgG (1:400; Molecular Probes) and Texas red goat anti-mouse IgG (1:400; Molecular Probes). The slides were examined with a Zeiss LSM780 Observer Z1 Confocal Laser Microscope or a Zeiss Axioplan 2E, motorized focus drive with a Hamamatsu monochrome digital camera. Images were acquired and processed by ZEN 2009 Light Edition and Adobe Photoshop software, and assembled in Adobe Illustrator (Adobe Systems).

## Transmission electron microscopy (TEM)

Negative staining and TEM were carried out as described previously (Snell, 1976) with slight modifications. Briefly, cells or ciliary ecotosomes in N-free medium were applied to 200 mesh carbon-formvar coated grids (Electron Microscopy Sciences, USA) and allowed to adhere for 30 s. The liquid was removed from the side by wicking with filter paper and the grids were washed twice with distilled water. Uranyl acetate (2%) was then applied for 10 s, and removed with filter paper. The samples were imaged on a FEI Tecnai G2 Spirit BioTWIN Transmission Electron Microscope. Images were processed by Adobe Photoshop software, and assembled in Adobe Illustrator (Adobe Systems).

## Acknowledgements

We are grateful to our colleagues in the Electron Microscopy Core Facility of UT Southwestern Medical Center at Dallas. This work was supported by National Institutes of Health Grants GM32843 to SKD and GM25661 and GM56778 to WJS.

## Additional information

### Funding

| Funder | Grant reference number | Author |
| --- | --- | --- |
| National Institute of General Medical Sciences (NIGMS) | GM25661 | William J Snell |
| National Institute of General Medical Sciences (NIGMS) | GM32843 | Susan K Dutcher |
| National Institute of General Medical Sciences (NIGMS) | GM56778 | William J Snell |

The funder had no role in study design, data collection and interpretation, or the decision to submit the work for publication.

### Author contributions

MC, JN, CIH-L, OB, QW, SKD, YL, WJS, Conception and design, Acquisition of data, Analysis and interpretation of data, Drafting or revising the article

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
