## [Decision Letter]

Thank you for sending your work entitled “Unidirectional ciliary membrane protein trafficking by the cytoplasmic retrograde IFT motor and cilia ectosome shedding” for consideration at *eLife*. Your article has been favorably evaluated by Randy Schekman (Senior editor) and 2 reviewers, including Maxence Nachury who is serving as a Guest Reviewing Editor.

The Reviewing editor and the other reviewer discussed their comments before we reached this decision, and the Reviewing editor has assembled the following comments to help you prepare a revised submission.

Both reviewers felt that the manuscript is an important contribution to ciliary signaling and trafficking, with very high disease significance given the established link between ciliary signaling and ciliopathies. Mechanistically, the data in Figure 4 are remarkable as they show that ectosomes contain a very specific set of proteins that is different from the protein complement of the cilium. This strongly suggests that specific proteins such as SAG are selected by a specialized machinery to become incorporated into forming ectosomes.

The logical progression of the experiments provides clarity and impact to the paper and the reviewers found little to fault in your submission. The major points that need to be addressed are as follow:

1) The conclusion that the IFT Dynein is active in the cytoplasm of *Chlamydomonas* is entirely reliant on the drug Ciliobrevin D. As this drug is a relatively newly developed reagent, further work may uncover caveats with its use such as off-target effects. The use of a temperature-sensitive mutant of IFT Dynein (described in [10]; doi:10.1083/jcb.201206068) to repeat the major findings (Figure 1) would greatly strengthen this part of the paper.

2) Several experiments need to be quantified to strengthen the main conclusions.

In particular, Figure 3 needs to be quantified to answer the question: “Does all of ciliary SAG1-C65-HA end up in the supernatant?”

In Figures 4 and 5: How many vesicles are found per cilium?

Figure 4: What is the enrichment factor of SAG, FMG1 and PKG in ectosomes? The abundance of each protein could be expressed as a relative western blot intensity (using infrared laser-based scanning of fluorescent secondary antibodies in the linear range of measurement) per mg of proteins in the sample.

---

## [Author Response]

*The logical progression of the experiments provides clarity and impact to the paper and the reviewers found little to fault in your submission. The major points that need to be addressed are as follow*:

*1) The conclusion that the IFT Dynein is active in the cytoplasm of* Chlamydomonas *is entirely reliant on the drug Ciliobrevin D. As this drug is a relatively newly developed reagent, further work may uncover caveats with its use such as off-target effects. The use of a temperature-sensitive mutant of IFT Dynein (described in*
[10]*; doi:10.1083/jcb.201206068) to repeat the major findings (*Figure 1*) would greatly strengthen this part of the paper*.

As shown in the revised manuscript, new experiments with an IFT dynein mutant confirmed the Ciliobrevin D results that the retrograde IFT motor is required for apical localization and ciliary entry of SAG1-C65-HA during gamete activation. Last summer we had initiated a collaboration with Dr. Susan Dutcher to introduce SAG1-HA into her dynein heavy chain 1b (DHC1b) temperature sensitive mutant, *fla24* (Lin et al., Cilia, 2:14, 2013), but we did not have the results until recently. Because we had those strains in hand, we have used them for the experiments in the revised manuscript instead of the *dhc1b-3* ts mutant described in [10], that was suggested in your letter. The phenotype of *fla24* is quite similar to that of *dhc1b-3*. The levels of DHC1b and the light intermediate chain of cytoplasmic dynein 1b, D1bLIC, are substantively depleted in both *fla24* and *dhc1b-3*, even at the permissive temperature. The extent of depletion was not quantified by Engels et al., but the amount of depletion shown in their blots appeared to be similar to that of Lin et al., who reported a 16-fold reduction compared to wild type of both proteins at the permissive temperature. Thus, both “ts” mutations in DHC1b lead to significant instability and degradation of the protein at the “permissive temperature”, and we carried out our experiments at the permissive temperature. As we indicate in the manuscript, the resting *SAG1-HAfla24* gametes have relatively large amounts of SAG1-C65-HA in their cilia at the permissive temperature compared to wild type, indicating that the retrograde IFT motor functions in membrane protein trafficking during the cilia formation that accompanies gametogenesis and possibly during ciliary maintenance in resting gametes. Importantly, though, SAG1-C65-HA apical redistribution and ciliary entry upon db-cAMP activation were blocked in the DHC1b-depleted *fla24* gametes.

*2) Several experiments need to be quantified to strengthen the main conclusions*.

*In particular,*
Figure 3
*needs to be quantified to answer the question*: *“Does all of ciliary SAG1-C65-HA end up in the supernatant?”*

*In*
Figures 4 and 5: *How many vesicles are found per cilium?*

Figure 4*: What is the enrichment factor of SAG, FMG1 and PKG in ectosomes? The abundance of each protein could be expressed as a relative western blot intensity (using infrared laser-based scanning of fluorescent secondary antibodies in the linear range of measurement) per mg of proteins in the sample*.

The revised manuscript now includes quantification of recovery of SAG1-C65-HA in the supernatant (75% recovered) and protein enrichment/de-enrichment in ciiary ectosomes (nearly 6-fold enrichment of SAG1-C65-HA and 0.1-fold of FMG1 in ciliary ectosomes compared to cilia). We used conventional blotting methods and ImageJ for our analysis under conditions in which the amount of protein loaded was proportional to the band intensities. We have omitted the PKG results because of a higher molecular mass, cross-reactive band of unknown origin that interfered with our analysis. We feel it would take a substantial amount of time to determine whether the higher mw band is a multimer of PKG or an artifact. Since the blots show that several ciliary proteins are substantially decreased and we quantify FMG1 de-enrichment, we do not feel that omission of PKG changes the interpretation or conclusions. We agree that it might be interesting to use TEM to quantify amounts of vesicles associated with cilia during release and when we add them back to cells. These are highly dynamic processes, and to avoid variation, we took samples at consistent times. Of course, there was some variability from experiment to experiment, but the TEM experiments were carried out multiple times by two of us, and the electron micrographs in Figures 4 and 5 are representative of typical amounts of vesicles we observed. To obtain full information about vesicles per cilium would require an extensive analysis at many time points, and, with the variability’s inherent in using unfixed samples and negative staining, we are just not sure that such an analysis would add important new information not already provided by the existing TEM and by our biochemical analysis.